# SIGLEC-1 in Systemic Sclerosis: A Useful Biomarker for Differential Diagnosis

**DOI:** 10.3390/ph15101198

**Published:** 2022-09-28

**Authors:** Jakob Höppner, Vincent Casteleyn, Robert Biesen, Thomas Rose, Wolfram Windisch, Gerd Rüdiger Burmester, Elise Siegert

**Affiliations:** 1Department of Rheumatology and Clinical Immunology, Charité—Universitätsmedizin Berlin, 10117 Berlin, Germany; 2Department of Pulmonology, Cologne Merheim Hospital, Kliniken der Stadt Köln gGmbH, Witten/Herdecke University, 51067 Cologne, Germany; 3Berlin Institute of Health at Charité—Universitätsmedizin Berlin, Charitéplatz 1, 10117 Berlin, Germany

**Keywords:** Systemic Sclerosis, SIGLEC-1, biomarker, interferon, treatment, cytokines

## Abstract

Systemic Sclerosis (SSc) is a clinically heterogeneous disease that includes an upregulation of type I interferons (IFNs). The aim of this observational study was to investigate the IFN-regulated protein Sialic Acid–Binding Ig-like Lectin 1 (SIGLEC-1) as a biomarker for disease phenotype, therapeutic response, and differential diagnosis in SSc. Levels of SIGLEC-1 expression on monocytes of 203 SSc patients were determined in a cross-sectional and longitudinal analysis using multicolor flow cytometry, then compared to 119 patients with other rheumatic diseases and 13 healthy controls. SSc patients higher SIGLEC-1 expression on monocytes (2097.94 ± 2134.39) than HCs (1167.45 ± 380.93; *p* = 0.49), but significantly lower levels than SLE (8761.66 ± 8325.74; *p* < 0.001) and MCTD (6414.50 ± 1846.55; *p* < 0.001) patients. A positive SIGELC-1 signature was associated with reduced forced expiratory volume (*p* = 0.007); however, we were unable to find an association with fibrotic or vascular disease manifestations. SIGLEC-1 remained stable over time and was independent of changes in immunosuppressive therapy. However, SIGLEC-1 is suitable for differentiating SSc from other connective tissue diseases. SIGLEC-1 expression on monocytes can be useful in the differential diagnosis of connective tissue disease but not as a biomarker for SSc disease manifestations or activity.

## 1. Introduction

Systemic Sclerosis (SSc) is a rare connective tissue disease that is characterized by the triad of microangiopathy, fibrotic complications and immunological abnormalities that include both innate and adaptive immunity [1,2,3]. One of the autoimmune phenomena is the production of characteristic and distinct serum autoantibodies detected in most patients as well as the presence of inflammatory cells with a prominent type I interferon (IFN) signature in circulating and tissue-infiltrating immune cells [4,5,6,7,8,9,10,11,12,13].

Activation of the type I IFN pathway is present in several rheumatic diseases including systemic lupus erythematosus (SLE), primary Sjögren syndrome (pSS), rheumatoid arthritis (RA), and others. While the direct detection of IFNs in plasma using ELISA is problematic and unreliable, several previous studies have attempted to establish indirect interferon markers, such as IFN regulated proteins, as biomarkers indirect IFN-markers in rheumatic diseases [8,9,10,11,12,13]. As sialic acid binding Ig like lectin 1 (SIGLEC-1), an IFN-induced adhesion molecule on monocytes [14], is one of the most prominent type I IFN-regulated genes, it has been the most promising marker so far. In pSS, SIGLEC-1 expression on peripheral blood monocytes could characterize patients with extraglandular involvement and high disease activity [15]. For myositis, SIGLEC-1 was found to be a candidate biomarker to assess type I IFN activity. It proved useful for monitoring disease activity and response to treatment in juvenile and adult dermatomyositis [16,17]. In addition, SIGLEC-1 has been shown to be elevated in RA [18], autoimmune thyroiditis [19], and primary biliary cholangitis (PBC) [20].

The most extensive data on the robustness of SIGLEC-1 as a biomarker for disease activity so far exists for SLE [21]. Biesen et al. were able to show that the frequency of SIGLEC-1-expressing monocytes correlates with disease activity and was inversely correlated with levels of complement factors. Moreover, glucocorticoid treatment resulted in a reduction in SIGLEC-1 expression in cells from adult patients with active SLE [22]. In addition, SIGLEC-1 expression was found to be a sensitive biomarker for adjusting disease activity in childhood SLE [23], and it has prognostic value for identifying SLE patients at risk for developing renal complications [13].

For SSc, the data is less clear. York et al. showed that IFN could induce SIGLEC-1 expression in SSc monocytes [10], and Farina et al. were able to show that SIGLEC-1 RNA expression in skin biopsies taken from fibrotic skin correlates with modified Rodnan skin score (mRSS) [24]. Moreover, Eloranta et al. found an association for IFNa and the interferon-inducible protein-10 (IP-10) in sera of SSc patients with cardiac involvement, signs of PAH, and a history of digital ulcers [25]. However, York et al. [10] and others were previously unable to demonstrate any differences with regard to skin involvement or organ complications in SSc patients for SIGLEC-1 expression on monocytes or soluble SIGLEC-1 in patient serum, respectively [13,26].

A further complication in SSc is that activity scores are poorly validated or can only be applied to specific subgroups, e.g., dcSSc [27]. Accordingly, it has been notoriously difficult to find appropriate biomarkers. Ideally, biomarkers that indicate general disease activity or specific organ manifestations, or that predict therapeutic response, would also be of great use in clinical practice.

The objective of the present study was to assess whether the expression of SIGLEC-1 on CD14^+^ cells via flow cytometry could serve as a useful biomarker for disease manifestation, including pulmonary or vascular complications and therapeutic response in SSc.

## 2. Results

### 2.1. Patients

203 SSc patients, 32 SLE, 16 pSS, 8 MCTD, 26 IIM, 14 UCTD, 23 RA, and 13 HCs were included in this study. Demographic data are shown in Table 1. Our SSc cohort was representative of the skewed proportion between females and males (84%/16%), as well as the proportions of patients with limited or diffuse cutaneous SSc and the age profile (46.67 ± 14.80 years at diagnosis) typical for Caucasian SSc patients (Table 2) [28]. A total of 115 SSc patients (56.7%) received immunosuppression, while 88 SSc patients were without immunosuppressive therapy. In addition, 28.9% of SSc patients on immunosuppressive therapy received hydroxychloroquine (in combination or alone). Comprehensive laboratory results were available for 97% of all SSc patients, pulmonary function test results were available for 83%, and echocardiography results for 60%.

As expected, SLE patients were slightly younger and pSS patients slightly older, which is in accordance with the expected age at disease onset for these conditions. Similarly, disease duration for UCTD patients is short, as many of them will later develop a distinct connective tissue disease.

### 2.2. SIGLEC-1 Expression in SSc and Control Groups

Comparing SIGLEC-1 expression on CD14^+^ monocytes in the peripheral blood of SSc patients to HCs, nearly half of SSc patients (47.8%) had monocyte SIGLEC-1 expression which was barely above the level of the HCs. Statistically, the expression of SIGLEC-1 (molecules/monocyte) was not significantly increased in SSc patients compared with HCs (2097.94 ± 2134.39 vs. 1167.45 ± 380.93, *p* = 0.49; Figure 1A).

In SSc patients, there was no difference between those who were receiving immunosuppression and those who were not (2164.94 ± 2385.49 vs. 1962.54 ± 1417.75, *p* = 0.47). In patients receiving hydroxychloroquine, there were also no differences when compared with those taking other immunosuppressive medications or with all SSc patients (with and without immunosuppression) (1510.60 ± 1157.08 vs. 2466.24 ± 2754.94, *p* = 0.34 and 1510.60 ± 1157.08 vs. 2358.18 ± 2406.02, *p* = 0.24).

When compared to other connective tissue diseases (CTDs) SIGLEC-1 expression was highest in SLE (8761.66 ± 8325.74), followed by MCTD (6414.50 ± 1846.55) and pSS (4371.69 ± 4227.89). RA (1425.22 ± 1312.69) and UCTD (1826.00 ± 1051.36) patients showed no elevated SIGLEC-1 expression when compared to HCs (1167.45 ± 380.93). Defining positive SIGLEC-1 expression as more than 2400 SIGLEC-1 molecules/monocyte, 45/203 (21.0%) SSc patients, 19/32 (59.4%) SLE, 8/16 (50.0%) pSS, 8/8 (100%) MCTD, 7/26 (26.9%) IIM, 4/14 (28.6%) UCTD, 1/23 (7.7%) RA, and 0/13 (0.0%) HCs had increased SIGLEC-1 levels. There was no correlation with either disease duration or age (r 0.007, r^2^ 0.00 and r 0.11, r^2^ 0.01) in SSc patients.

### 2.3. SIGLEC-1 Expression and SSc Manifestations

When comparing SIGLEC-1 expression of SSc patients according to the different organ manifestations, SIGLEC-1 positive SSc patients showed significantly impaired forced vital capacity (FVC) (81.39 ± 18.67 vs. 91.34 ± 20.09; *p* = 0.007); however, no differences were found regarding the prevalence of ILD (46.7% vs. 44.3%; *p* = 0.779), and there was no difference in absolute SIGLEC-1 expression between patients with and without ILD (2068.89 ± 1963.12 vs. 2129.13 ± 2266.90; *p* = 0.427) (Tab 2 and Figure 1B). When analyzing the different SSc manifestations according to SIGLEC-1 positivity, it could be found in 23/122 (18.9%) lcSSc patients; 17/64 (25.0%) dcSSc patients; 21/91 (19.8%) patients with ILD; 4/19 (21.1%) PAH patients; 26/111 (22.5%) patients with vascular complications including PAH, DU, and SRC, 21/97 (20.6%); and 2/10 (20.0%) patients with myositis (Table 2). SILGEC-1 positive patients tended to a higher prevalence of cardiac involvement (11.1% vs. 4.4%; *p* = 0.094) and a reduced left ventricular ejection fraction (LVEF) (59.44 ± 11.22 vs. 63.03 ± 9.08; *p* = 0.098).

### 2.4. SIGLEC-1 Expression and SSc-Specific Autoantibodies

In U1RNP positive SSc patients, SIGLEC-1 expression was strongly increased (9055.04 ± 6862.59) compared to other SSc autoantibodies (*p* = 0.003). Moreover, compared to all SSc patients, RP3 positive patients tended to have increased SIGLEC-1 expression (3376.94 ± 3821.81 vs. 1984.86 ± 1899.83, *p* = 0.136) (Figure 1C). Interestingly, this group showed a significantly higher mRSS compared to other SSc patients (13.38 ± 8.35 vs. 5.86 ± 6.70, *p* = 0.003).

### 2.5. Association of SIGLEC-1 Expression and mRSS

When investigating all SSc patients included in our study, no correlation was detected between the level of SIGLEC-1 expression and skin involvement quantified by mRSS using a linear regression model (r^2^ 0.01 and r 0.09; Figure 1D). As mentioned above, there was no significant difference in SIGLEC-1 expression between dcSSc patients and lcSSc patients (2430.48 ± 2384.46 vs. 1959.05 ± 2089.78, *p* = 0.16).

### 2.6. Longitudinal SIGLEC-1 Expression in Treated and Untreated SSc Patients

For 62 SSc patients, follow-up SIGLEC-1 measurements were available after an average of 281 days (±175 days). The median change in SIGLEC-1 expression was 0.00 (interquartile range [IQR] 962.00). The vast majority (*n* = 49; 79.0%) of individuals with available longitudinal samples remained in their respective SIGLEC-1 high or low category over that follow-up period (Figure 2). Of the remaining patients, three increased from negative SIGLEC-1 to positive and ten patients changed vice versa. There were no significant changes in SIGLEC-1 expression over the follow-up period. This was also true for patients with constant immunosuppression and for patients without immunosuppression during follow-up (paired *t*-test *p* = 0.30 and *p* = 0.05; Figure 2A,B). Similarly, there was no effect of the change in therapy on SIGLEC-1 expression. Thus, no significant changes were found in either the case of escalation or de-escalation of immunosuppressive therapy (paired *t*-test *p* = 0.48 and *p* = 0.57; Figure 2C,D).

### 2.7. SIGLEC-1 as Biomarker in Differential Diagnosis of SSc

SIGLEC-1 expression was significantly increased in SLE and MCTD patients when compared with SSc (8761.66 ± 8325.74 vs. 2097.94 ± 2134.39; *p* < 0.0001 and 6414.50 ± 1846.55 vs. 2097.94 ± 2134.39; *p* = 0.0003). ROC analysis revealed a SIGLEC-1 expression of 4806 molecules/monocyte as optimal median cut point to differentiate SSc from SLE (sensitivity 93.1%, specificity 50.0%, area under the curve [AUC] = 0.76, Youden’s index 0.43; Figure 3A), 3303 molecules/monocyte as optimal cut point to differentiate SSc from MCTD (sensitivity 85.71%, specificity 100%, AUC = 0.95, Youden’s index 0.86; Figure 3B), and 3768 molecules/monocyte as optimal cut point to differentiate SSc from SLE or MCTD (sensitivity 87.68%, specificity 62.5%, AUC = 0.80, Youden’s index 0.50; Figure 3C). Using our previously established cut-off value of more than 2400 SIGLEC-1 molecules/monocyte to distinguish between positive and negative SIGELC-1 expression, the sensitivity to distinguish SSc from SLE is 78.33% and the specificity is 59.38% (AUC = 0.76, Youden’s index 0.38); to distinguish SSc from MCTD, the sensitivity is 78.33% and the specificity is 100% (AUC = 0.95, Youden’s index 0.78); and to distinguish SSc from SLE or MCTD, the sensitivity is 78.33% and the specificity is 67.5% (AUC = 0.80, Youden’s index 0.46).

## 3. Discussion

Over the last few decades, growing evidence suggesting activation of type I IFNs and their pathways in the pathogenesis of SSc has emerged [5,29,30,31,32]. Specifically, it was shown that SIGLEC-1 is upregulated both on SSc monocytes and on tissue macrophages [10,22]. Indeed, we could find a trend for elevated SIGLEC-1 expression on monocytes in SSc patients when compared to healthy controls, although we could not find statistical significance. However, this elevation was markedly lower than the one seen in other CTDs. In our study, a positive SIGELC-1 signature was associated with a reduced FVC; however, we did not observe an association with ILD. In addition, patients with a positive SIGELC-1 signature tended to have a higher prevalence of cardiac involvement alongside with a reduced LVEF. Our data regarding cardiac involvement fit previous results by Eloranta et al. [25]. However, we were unable to identify further associations. Moreover, we were unable to detect major changes in expression levels over time. A lack of association between SIGLEC-1 level and clinical phenotype has previously been reported by others within smaller cohorts [10,13]. Notably, unlike in dermatomyositis, we were also unable to detect any type-I IFN signature in muscle biopsies of SSc patients [33].

Evaluating the use of SIGLEC-1 as a marker of response to therapy, we found that SIGLEC-1 expression is largely independent of changes in immunosuppression in SSc patients. This is in contrast with previous findings in SLE or pSS, where an effect of immunosuppressive therapy on SIGLEC-1 expression could be seen [15,22]. We did not see any difference regarding SIGLEC-1 levels between patients receiving immunosuppressive treatment and patients who did not, including patients receiving hydroxychloroquine. In fact, hydroxychloroquine blocks Toll-like receptors (TLR) 7 and 9, and was shown to inhibit type I IFN production in SLE [34]. In addition, it was shown in pSS that hydroxychloroquine significantly reduces SIGLEC-1 expression [15]. In our cohort, SIGLEC-1 expression remained largely constant over time in SSc patients, even with increases or decreases in immunosuppressive therapy, including hydroxychloroquine or other drugs such as glucocorticoids, methotrexate, and rituximab which are known to decrease type-I IFN production.

Despite the fact that SIGLEC-1 expression on monocytes might not be a good biomarker in SSc, there are several reports that have demonstrated an upregulation of IFN-regulated proteins, including SIGLEC-1, in SSc skin and other organs affected by fibrotic complications. In biopsy studies, tissue expression of SIGLEC-1 correlated with mRSS [24]. Moreover, microarray analysis of lung tissues derived from SSc patients revealed that expression of IRGs correlated with progressive ILD [35].

This finding is particularly relevant, as there are now several immunosuppressive medications that target IFN pathways, such as anifrolumab [36,37] and JAK inhibitors [38]. These new medications have shown great potential in the therapy of RA and SLE, and it has been shown that the baseline type I IFN signature predicts the response to anifrolumab therapy in SLE [36]. Interestingly, JAK inhibitors are effective in RA despite relatively low SIGLEC-1 expression [39]. Similarly, there are initial reports suggesting that JAK inhibitors and anifrolumab might have beneficial effects on both vascular and fibrotic manifestations of SSc [40,41,42].

Our negative findings might reflect the fact that we used a rather insensitive method to determine type-1 IFN activation. Firstly, the determination of several IRGs might be a more sensitive approach. Secondly, quantitative PCR might be more sensitive than the quantitative assessment of a single protein by immunohistochemistry, ELISA, or flow cytometry. Lastly, there might be a discrepancy between local, tissue-specific findings and systemic findings on peripheral blood cells. In line with this, Hesselstrand et al. showed in SSc that the plasma IFN signature remains relatively constant during paquinimod therapy, while the IFN signature in the skin decreases [43].

Limitations of this study include the fact that disease activity in SSc was not assessed by score, and thus there was a lack of correlation between disease activity in SSc and SIGLEC-1 expression. As mentioned earlier, reliable assessment of disease activity in SSc is a major challenge due to the lack of good established tools and scores. We therefore used disease complications as a parameter for disease severity. Nevertheless, even these complications show a great heterogeneity, and the significance must be interpreted with caution. In our cohort, we could not find any effect of immunosuppressive therapy. However, these data were collected in routine clinical practice and were not prospectively evaluated according to a fixed protocol. Another limitation is the small number of patients who were analyzed with newly diagnosed SSc. In these patients with very short disease duration, the benefits of SIGLEC-1 would be interesting and should be investigated in future studies. Nevertheless, we could not find a correlation between SIGLEC-1 and disease duration in the present data. Finally, it should be noted that our results refer to the systemic measurement of SIGELC-1 in peripheral blood. Other data suggest that IFN markers in tissue may indeed have prognostic value.

Another potential role for SIGLEC-1 expression on monocytes is that of facilitating differential diagnoses. There is sometimes a challenge as to whether a patient with a suspected connective tissue disease has or develops SSc or SLE [44]. In this scenario, the fact that patients with SLE or MCTD showed significantly increased SIGLEC-1 expressions compared with patients with SSc can be used in combination with patients’ clinical presentation, as well as the autoantibody profile, to guide early differential diagnosis. As ROC analysis showed, markedly increased SIGLEC-1 expression makes the diagnosis of SLE or MCTD much more likely. Moreover, clinical characteristics of CTD patients may evolve over time, which may result in a “clinical shift” from MCTD to another CTD such as SLE or SSc [45]. We demonstrated that MCTD and SLE patients showed significantly higher SIGLEC-1 expression than SSc patients. These findings fit with previous descriptions of the IFN signature in MCTD patients [4]. Hence, SIGLEC-1 expression could be a useful biomarker to attribute a patient with early and unspecific disease manifestations to a certain phenotype, and then base treatment decisions on this immunological information. Recently, Zorn-Pauly et al. investigated SIGLEC-1 in patients with suspected SLE and revealed that a negative test result for SIGLEC-1 is able to exclude SLE in suspected cases [46]. These data support our approach of using SIGLEC-1 expression to distinguish SSc from other CTDs such as SLE and MCTD.

The key findings of our study are that SIGLEC-1 expression on monocytes is mildly, but not significantly, elevated compared to healthy controls. We were also unable to find clear associations with clinical manifestations or with changes in immunosuppressive therapy. However, SIGLEC-1 expression may be valuable in differentiating SSc from MCTD or SLE, and showed a clear association with anti-U1RNP antibodies.

## 4. Material and Methods

### 4.1. Study Design

For this observational longitudinal study, patients and healthy controls from our center at the Department of Rheumatology, Charité—Universitätsmedizin Berlin, Germany were recruited. The study protocol was approved by the Charité—Universitätsmedizin Berlin Ethics Committee (EA1/179/17). Written informed consent was obtained from each patient. The study was conducted in accordance with the principles of the Declaration of Helsinki.

### 4.2. Patients

Patients were included in the study if they agreed to participate and were affected by one of the following rheumatic conditions: SSc, SLE, pSS, mixed connective tissue disease (MCTD), idiopathic inflammatory myositis (IIM), undifferentiated connective tissue disease (UCTD), or RA, and met the respective diagnostic or classification criteria for their rheumatic disease, or if they were healthy without any evidence of acute infection or chronic disease (HC = healthy controls). Diagnostic and classification criteria used were 2013 ACR/EULAR classification criteria for SSc [47], 2019 EULAR/ACR Classification Criteria for SLE [48], and 2016 ACR-EULAR classification criteria for pSS [49]. MCTD was diagnosed according to Alarcon-Segovia et al. [50], 2017 EULAR/ACR for IMM [51] and 2010 ACR/EULAR criteria for RA [52].

Demographic, clinical, and serological data were collected according to standardized procedures. For SSc patients, this included cutaneous subsets, age at onset of Raynaud’s phenomenon, age at onset of first non-Raynaud’s phenomenon symptom, disease duration, organ involvement, and immunosuppressive therapy at the time the blood samples were taken. Other variables collected included smoking history, digital ulcers (DU), calcinosis, highest mRSS, systemic hypertension, hyperlipidemia, diabetes mellitus, myocardial infarction, angina pectoris, stroke, transitory ischemic attack (TIA), periphery arterial disease (PAD), PAH, interstitial lung disease (ILD), scleroderma renal crisis (SRC), heart involvement, and myositis. Laboratory parameters (C-reactive protein [CRP], neutrophile count, hemoglobin, and N-terminal pro-B-type natriuretic peptide [NT-proBNP]) were quantified from peripheral blood during clinical routines. Lung function was assessed via spirometry. Diffusing capacity for carbon monoxide (DLCO) was measured using the single-breath method. Spirometry and echocardiography were performed on all patients as part of the annual examinations. If the presence of ILD or PAH was clinically suspected or indicated by these examinations, high-resolution computed tomography (HRCT) or a right-sided heart catheterization was performed. PAH was defined as a mean pulmonary artery pressure of ≥ 25 mmHg and a pulmonary capillary wedge pressure of ≤ 15 mmHg on right-sided heart catheterization. ILD was defined as the presence of pulmonary fibrosis on a high-resolution computed tomography scan evaluated by experienced radiologists. For patients who presented to our centre several times, these data were collected again as part of a follow-up visit.

### 4.3. Multi-Color Flow Cytometry for SIGLEC-1 Validation

SIGLEC-1 on CD14-positive monocytes was measured as described previously [39]. In brief, EDTA-anticoagulated whole blood was incubated with 10 mL of mouse-anti-human antibody cocktail containing phycoerythrin (PE)-labeled anti-CD169 monoclonal antibody (mAb) (labeled with a fluorochrome/protein ratio of 1:1), allophycocyanine (APC)-labeled anti-CD14 mAb and Krome Orange-labeled anti-CD45 mAb (all antibodies from Beckman Coulter, Krefeld, Germany). Red blood cells were then lysed by addition of 500 mL of Versa-Lysis solution (Beckman Coulter) to each reaction tube. After incubation, samples were centrifuged. Samples were then washed, acquired on a 10-color flow cytometer, and centrifuged again. They were then stained (Navios, Beckman Coulter) and analyzed using the Navios software. During each analytical run, QuantiBRITE TM PE tubes (BD Biosciences) were used to convert the fluorescent channel 2 (FL2) mean fluorescent intensity (MFI) signals on CD14^+^ monocytes to monoclonal antibodies bound per cell (mAb/cell) values. FL2 MFI values and absolute values of PE molecules (as given by the manufacturer) for each QuantiBRITE TM bead population were used to perform linear least square regression analysis in order to determine the best calibration value. This was then used to convert the FL2 MFI values of monocytes in the analytical sample into the amount of PE-labeled CD169 mAb bound per monocyte (mAb/monocyte). The reference range for the expression of SIGLEC-1 in healthy controls was determined to be less than 2400 SIGLEC1 molecules/monocyte. SIGLEC-1 expression was assessed via flow cytometry with a detection limit of 1200 molecules/monocyte. Values below the limit of detection (LOD) are shown as LOD/√2.

### 4.4. Statistical Analysis

Statistical analysis was performed by using Jamovi version 2.3 1.6 for Mac (the jamovi project, 2021), retrieved from https://www.jamovi.org (accessed on 6 September 2022) and GraphPad Prism version 8.4.3 for Mac (GraphPad Software, San Diego, CA, USA). Data are presented as mean ± standard deviation (SD) of mean if not otherwise indicated. Data were tested for normal distribution using the Shapiro-Wilk test. The Mann–Whitney U test (non-parametric) or the unpaired t-test (parametric) was performed for continuously distributed variables for the purpose of comparison between the two groups. For the follow-up data, the paired t-test was used, as well as the Wilcoxon rank test for validation. For comparison of more than two groups, data were analyzed by one-way ANOVA (parametric) followed by Dunnett’s multiple comparisons test or Kruskal–Wallis test (non-parametric) followed by Dunn’s multiple comparisons test. For categorical variables, either the chi-square test or Fisher’s exact test was performed. A *p*-value of <0.05 was considered statistically significant. Receiver operating characteristic (ROC) analysis was performed to define cut off values.

## 5. Conclusions

We demonstrated in a large cohort that patients with SSc show a slightly elevated SIGLEC-1 expression on monocytes compared to healthy controls, but SIGLEC-1 expression was much lower compared to other CTDs, such as SLE and MCTD. Our data on the use of SIGLEC-1 expression on monocytes as a marker for organ manifestations remain ambiguous, as we did not find any evidence supporting the use of SIGLEC-1 as a biomarker for disease activity or response to therapy in SSc. SIGLEC-1 expression levels remained largely constant during disease progression and were not significantly affected by changes in therapy. On the other hand, we found that SIGLEC-1 is valuable for the early differential diagnosis of SSc and may be helpful in distinguishing SSc from SLE or MCTD.

## Figures and Tables

**Figure 1 pharmaceuticals-15-01198-f001:**
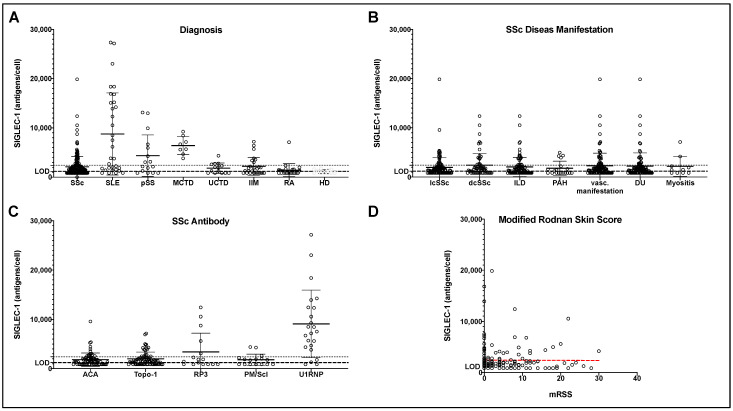
Expression analysis of sialic acid-binding immunoglobulin-like lectin 1 (CD169/SIGLEC-1) on circulating monocytes. (**A**): In patients with SSc, SLE, SS, MCTD, UCTD, IIM, RA, and HDs; (**B**): SSc disease manifestations, including limited (lcSSc) and diffuse (dcSSc) skin involvment, interstitial lung disease (ILD), pulmonary arterial hypertension (PAH), vascular manifestations (referring to PAH, digital ulcera, and scleroderma renal crisis), digital ulcera (DU), and SSc-associated myositis; (**C**): SSc specific antibodies, including anti-centromer antibodies (ACA), anti-topoisomerase-1 antibodies (Topo-1), RNA-polymerase 3 antibodies (RP3), PM/Scl and U1RNP antibodies. (**D**): Correlation of modified Rodnan Skin Score (mRSS) and SIGELC-1. The dashed line indicates the lower detection limit of 1200 molecules/monocyte. Values below the limit of detection (LOD) are shown as LOD/√2. The dotted line indicates the reference range for SIGELC-1 expression.

**Figure 2 pharmaceuticals-15-01198-f002:**
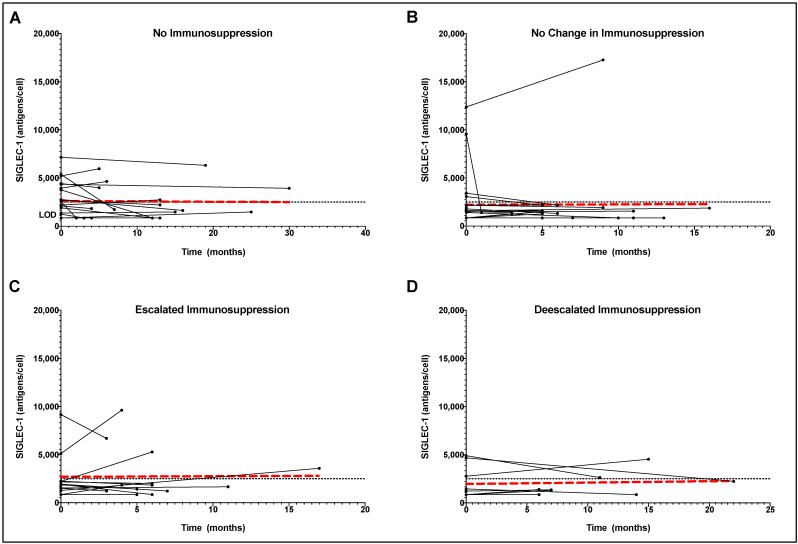
Longitudinal analysis of SIGELC-1 expression of Systemic Sclerosis (SSc) patients without receiving immunosuppression (**A**), with unchanged immunosuppression (**B**), and with escalated (**C**) or deescalated (**D**) immunosuppressive therapy. The red dashed line represents the trend of all patients in the respective SSc group.

**Figure 3 pharmaceuticals-15-01198-f003:**
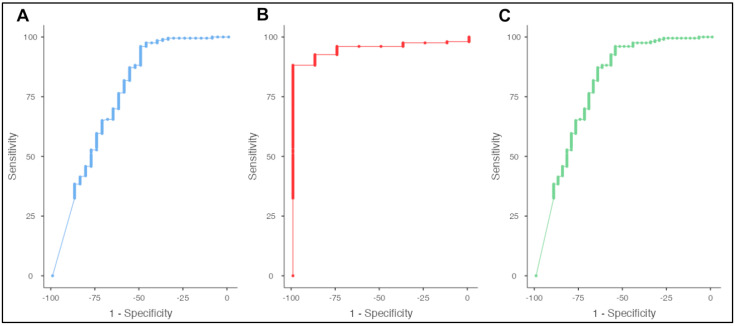
ROC curve analysis of SIGLEC-1 in SSc patients vs. SLE patients (**A**), SSc patients vs. MCTD patients (**B**), and SSc patients vs. SLE or MCTD patients (**C**).

**Table 1 pharmaceuticals-15-01198-t001:** Demographic characteristics.

		SSc	SLE	pSS	MCTD	UCTD	IIM	RA	HC
		*n* = 203	*n* = 32	*n* = 16	*n* = 8	*n* = 14	*n* = 26	*n* = 23	*n* = 13
**Sex**									
	No. (%) female	171 (84.2)	28 (87.5)	15 (93.8)	8 (100.0)	9 (64.3)	16 (61.5)	17 (73.9)	10 (76.9)
	No. (%) male	32 (15.8)	4 (12.5)	1 (6.2)	0 (0.0)	5 (35.7)	10 (38.5)	6 (26.1)	3 (23.1)
**Age** (mean ± SD; yrs)		57.84 ± 14.31	41.45 ± 12.70	66.00 ± 9.69	56.63 ± 19.15	58.93 ± 11.57	61.85 ± 14.42	62.44 ± 14.69	55.95 ± 15.49
	At diagnosis	46.67 ± 14.80	31.56 ± 12.31	56.75 ± 11.96	41.60 ± 23.36	58.00 ± 8.91	59.14 ± 15.29	51.83 ± 17.33	N/A
**Disease duration** (mean ± SD; yrs)		10.15 ± 8.81	8.83 ± 8.58	8.17 ± 6.45	9.80 ± 9.42	0.50 ± 0.58	3.00 ± 2.77	10.72 ± 12.02	N/A
**Antinuclear antibody** positive, no. (%)		186 (92.1%)	31 (96.9%)	15 (93.7)	8 (100%)	12 (85.5)	15 (57.7)	8 (34.8%)	2 (15.3)

HC = healthy control; IIM = idiopathic inflammatory myositis; MCTD = mixed connective tissue disease; pSS = primary Sjögren’s Syndrome; RA = rheumatoid arthritis; SD = standard deviation; SLE = Systemic lupus erythematosus; SSc = Systemic Sclerosis; UCTD = undifferentiated connective tissue disease; yrs = years; disease duration refers to the time since first non-Raynaud symptom in SSc.

**Table 2 pharmaceuticals-15-01198-t002:** Clinical and serologic characteristics of SSc patients.

		All SSc Patients (*n* = 203)	Negative SIGELC-1 Signature (*n* = 158)	Positive SIGLEC-1 Signature (*n* = 45)	*p* Value
**Cutaneous subset**—*n* (%)					
	diffuse (dcSSc)	64 (31.5)	47 (29.7)	17 (37.8)	0.306
	limited (lcSSc)	122 (60.1)	99 (62.7)	23 (51.1)	0.163
	sine scleroderma (ssSSc)	17 (8.4)	12 (7.6)	5 (11.1)	0.453
**Immunological findings**					
	ANA	186 (94.4)	147 (93.0)	39 (86.7)	0.825
	ACA	71 (35.0)	58 (36.7)	13 (18.3)	0.332
	Topo-1	76 (37.4)	58 (36.7)	18 (28.9)	0.687
	RP3	16 (7.9)	11 (7.0)	5 (11.1)	0.339
**SSc organ manifestations**, *n* (%)					
	Raynaud’s phenomenon	181 (89.2)	144 (91.1)	37 (82.2)	0.090
	ILD	91 (44.8)	70 (44.3)	21 (46.7)	0.779
	PAH	19 (9.4)	15 (9.5)	4 (8.9)	0.902
	DU	97 (47.8)	76 (48.1)	21 (46.7)	0.865
	Cardiac involvement	12 (5.9)	7 (4.4)	5 (11.1)	0.094
	SRC	8 (3.9)	5 (3.2)	3 (6.7)	0.287
	Myositis	10 (4.9)	8 (5.1)	2 (4.4)	0.866
**Laboratory values** (mean ± SD)					
	NT-pro-BNP—ng/L	451.09 ± 1174.33	392.14 ± 947.07	646.29 ± 1726.65	0.204
	CRP—mg/dl	4.36 ± 9.70	4.00 ± 9.72	5.78 ± 9.60	0.279
	Hb—mg/dl	13.10 ± 1.62	13.10 ± 1.62	13.15 ± 166	0.815
	Neutrophil granulocytes	5.54 ± 2.63	5.66 ± 2.67	5.15 ± 2.46	0.275
**Cardiopulmonary parameters** (mean ± SD)					
	FVC—%/exp.	89.27 ± 20.16	91.34 ± 20.10	81.40 ± 18.70	0.007
	FEV1—%/exp.	86.30 ± 20.78	87.20 ± 21.53	83.00 ± 17.66	0.283
	DLCO—%/exp.	57.29 ± 18.91	58.29 ± 19.49	53.49 ± 16.19	0.195
	LVEF—%/exp.	62.27 ± 9.63	63.03 ± 9.08	59.44 ± 11.22	0.098

CRP, C-reactive protein; DLCO, diffusing capacity for carbon monoxide; FEV1, forced expiratory volume per second; FVC, forced vital capacity; Hb, hemoglobin; ILD, interstitial lung disease; L, liter; LVEF, left ventricular ejection fraction; *n*, number; NT-proBNP, N-terminal-pro-brain natriuretic peptide; PAH, pulmonary arterial hypertension; SRC, scleroderma renal crisis; %/exp, percent expected.

## Data Availability

Data is contained within the article.

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
