# Peer review of "SIGLEC-1 in Systemic Sclerosis: A Useful Biomarker for Differential Diagnosis"

_pharmaceuticals, 2022, doi:10.3390/ph15101198_

Round 1
Reviewer 1 Report
Introduction:
Page 2 para 2 - the word activates is repeated twice
Page 2 para 2 - JAK1 und TYK2 promotion the activation of IFN-stimulated genes (ISGs) via IFN-regulatory factor 9 - Please correct the language errors
Page 2 para 3 - correct spelling of thyroiditis
Page 2 para 5 - what is mRSS - Please ensure abbreviations are expanded at the instance of its first use
Introduction is too long and lacks focus. Please cut-it short, to the point and enable the reader to quickly get in to the topic of research.
Methods
Please add section on ethics - was the study approved by IEC. Was it done according to the declaration of Helsinki and other local rules and regulations. Add a line on informed consent
The STROBES guidelines has not been followed - a lot of items from the checklist is missing -please add all those sections
Statistics - Statistical analysis was performed by using Jamovi 2.3 and GraphPad Prism 8.4.3. - Please add the name of the company, headquarters and year as how these proprietary softwares are metnioned in a scientific paper
There is no mention of how parametric or non-parametric was assessed? What test of normality was used?
The statistical tests are not appropriate. There are more than two groups - SSc, SLE, pSS, MCTD, RA, UCTD, IIM HC. In that case, ANOVA or Kruskal wallis to be done and then post hoc test to be done to avoid multiple comparisons. When there are may groups in the study, taking only SSc and comparaing it with HC and giving a p value and ignoring all other groups as though they dont exist, is not appropriate is what I feel. The numbers of the HCs are also very less
For the longitudinal analysis - paired t test is used. Were all assumptions for paired t-test fulfilled?
Discussion and conclusion:
I am a little skeptical to call the SIGLEC-1 can be used in the differential diagnosis in connective tissue disorders. Without definitive cut off points, how will this be possible? In this study "SSc patients showed significantly higher SIGLEC-1 expression on PBMCs (2097.94 ± 2134.39) than HCs (1167.45 ± 380.93) but lower levels than SLE (8761.66 ± 8325.74) and MCTD (6414.50 ± 1846.55) patients". Now we have a patient whose GIlec level comes as 4300. Will we consider this patient as SSc and MCTD?
Author Response
We thank Reviewer 1 very much for the comments! In our eyes, our manuscript has benefited greatly from the appropriate revision.
For a detailed point-by-point response, please see the attachment.

Reviewer 2 Report
In this manuscript the authors measured expression of SIGLEC-1 on PBMCs from healthy controls and patients with connective tissue-diseases, focusing on a larger cohort of patients with SSc. They investigated expression levels between clinical entities, and between various disease manifestations and treatment response in SSc.
I found the manuscript to be well written over all, and the study carried out to a high standard.
I have a few minor points for the authors to address.
In the abstract it would be good to include the p-values of the associations described.
I found the introduction to be a little overly long, it could be made more concise.
Please ensure all abbreviations are explained the first time they are used, eg mRSS is used in the introduction but not expanded until the methods.
In the methods section on patients, patients with SSc are not actually mentioned, just all of the other CTDs included.
Were HRCT, right-heart catheterization, and spirometry etc performed in all SSc patients? There is no mention of missing data in the methods or in Table 2, but it is unlikely that all patients underwent all of these investigations.
In the result section on the patients included in the study there are 2 points:
- Presumably the description of the patients on treatment or not was at the time of the blood sample was taken, but this is not made clear.
-It is mentioned that the demographics of the cohort is typical for Caucasian SSc patients. There is however no other mention of ethnicity of the patients anywhere else. Were the patients all of European descent?
There is not always a p-value given for the group comparisons, eg in patients with DU.
The comparison between patients with lcSSc and dcSSc is slightly contradictory. It is stated that there was a trend to increased expression, but later it is stated that there was no difference, so this is a slightly confusing message.
Author Response
We thank Reviewer 2 very much for the comments! In our eyes, our manuscript has benefited greatly from the appropriate revision.
Please see the attachment for a detailed point-by-point response.

Round 2
Reviewer 1 Report
The authors have addressed all my currents in an excellent manner and I am happy the way the manuscript has currently come out. I thank the authors for taking the comments on a positive note and for having constructively worked on them to improve the science of the paper. I just have one small comment. Very minor
Tests of Normality usually is Shapiro Wilk for small sample sizes. Kolmogorov Smirnov is for large datasets like sample size > 2000
Author Response
Once again, we would like to thank Reviewer 1 for his very constructive comments!In our eyes, our manuscript has benefited greatly from them.
Regarding Reviewer 1‘s comment on the test for normality, we are very grateful for the information that the Shapiro-Wilk test is better suited for smaller samples. We have re-assessed our data using the Shapiro-Wilk test and arrived at the same results as before with the Kolmogorov Smirnov test. We have included the Shapiro-Wilk test in the methods section. Thank you very much for this important advice.